# Moving beyond Symptom Criteria to Diagnose and Treat Functional Disorders: Patient-Reported Symptoms of Functional Lower Gastrointestinal Disorders Correlate Poorly with Objective Assessment of Luminal Contents Seen on Intestinal Ultrasound

**DOI:** 10.3390/jcm13164759

**Published:** 2024-08-13

**Authors:** Claudia Brick, Heidi Su, Kirstin Taylor, Rebecca Burgell

**Affiliations:** 1Alfred Health, Melbourne, VIC 3004, Australia; c.brick@alfred.org.au (C.B.);; 2School of Translational Medicine, Monash University, Melbourne, VIC 3800, Australia

**Keywords:** disorder of brain–gut interaction, intestinal ultrasound, functional gastrointestinal disorder, irritable bowel syndrome

## Abstract

**Background/Objectives:** The diagnosis of lower functional gastrointestinal disorders (FGIDs) is currently based on subjective and unreliable patient-reported symptoms, with significant clinical overlap between diagnosed phenotypes. Objective biomarkers are urgently sought. Gastrointestinal ultrasound (GIUS) can objectively and non-invasively assess luminal contents. This study aimed to assess the utility of GIUS in phenotyping patients with lower FGIDs. **Methods**: Patients with lower FGIDs underwent a GIUS and completed the Rome IV Diagnostic Questionnaire, SAGIS questionnaire, and 100 mm VAS score for overall symptom severity. The faecal loading score (FLS) was obtained using a modified Leech score, where an FLS of >37 was consistent with clinically significant constipation. **Results**: Eighty-eight patients fulfilled the study requirements. In total, 56 met the Rome IV criteria for irritable bowel syndrome (IBS) subtypes, while 23 met the criteria for functional constipation (FC), 4 for functional diarrhoea (FD), and 5 for other diagnoses. Patients reporting constipation-predominant symptoms had a significantly higher median FLS than those describing diarrhoea-predominant symptoms (FLS = 40 [IQR 20.0–53.3] vs. 13.3 [IQR 6.7–40.0], respectively). However, 27% of patients describing diarrhoea had significant faecal loading on GIUS, and of those who described constipation, 34% did not have significant faecal loading. Sensitivity and specificity for the detection of FLS-indicated constipation by the Rome IV criteria were low at 59% and 66%, respectively. **Conclusions**: The symptom-based diagnosis of FGID subtypes based on the Rome IV criteria is a poor predictor of faecal loading. These findings should prompt further exploration of the limitations of symptom-based assessment and a shift towards physiological assessment of patients with FGIDs such as gastrointestinal ultrasound to develop more targeted therapy. Future research is underway to determine if targeting objective physiological endpoints results in improved clinical outcomes.

## 1. Introduction

The diagnosis of lower functional gastrointestinal disorders (FGIDs) relies predominantly on patient-reported symptoms, with the exclusion of other significant pathology with examination and simple investigations [1]. The Rome criteria, first established in 1989, are the gold standard for the diagnosis of these disorders of gut–brain interaction and classify patients into subcategories based on symptoms to guide treatment and further research [2,3]. Affecting up to 10% of adults worldwide with varying prevalence between countries, the hallmark symptoms of functional bowel disorders are altered bowel habits and abdominal pain [4]. Categories include irritable bowel syndrome (IBS) with predominant diarrhoea (IBS-D), IBS with constipation (IBS-C), IBS with a mixed bowel pattern (IBS-M), IBD unclassifiable (IBS-U), functional constipation (FC), and functional diarrhoea (FD) [5]. It is likely that underlying each symptom-based category, there are multiple contributing aetiologies that are poorly understood. In addition, symptoms may not correlate with underlying gastrointestinal physiological changes. For instance, overflow diarrhoea is a commonly recognised presentation in individuals with significant constipation. 

It is widely accepted that these diagnoses are not static, and there may be considerable overlap between categories [6,7]. The majority of patients experience both constipation and diarrhoea, and these symptoms may fluctuate multiple times per month [8,9]. Up to 75% of patients transition between subtypes over time. This generally occurs when patients with diarrhoea or constipation-predominant subtypes shift to a mixed pattern (IBS-M), whereas few completely switch from IBS-C to IBS-D or vice versa [10,11]. It follows that these diagnoses may exist on a spectrum rather than as discrete entities: Multiple studies have suggested that IBS-C and functional constipation are not separable [12,13,14,15]. Wong et al. demonstrated that if the Rome criteria requirement that diagnoses must be mutually exclusive was suspended, up to 90% of IBS-C patients would also fulfil the criteria for functional constipation, and at 1-year follow-up, two-thirds of patients had changed diagnosis [12]. Similarly, IBS-D and functional diarrhoea may be difficult to differentiate [16,17].

Subjective symptoms reported by patients are also known to be inaccurate. Recall of past symptoms has been demonstrated to be significantly different from a daily diarised recording, possibly because of the variability in stool form experienced over a period of time biases’ recall [18,19]. Patients’ definitions of diarrhoea and constipation may vary from that of the clinician, and patients are more likely to classify their symptoms based on form rather than frequency [20,21]. Supporting this, a recent study objectively analysed stool consistency to demonstrate that it correlated poorly with patients’ descriptions: Only 60% of IBS-D patients had diarrhoea on objective measures [22]. This unreliability of symptoms further confounds symptom-based diagnosis. 

The treatment of IBS remains reliant on these subjective and non-specific symptoms. Although there has been a rise in the popularity of non-pharmacological management strategies including the low FODMAP diet and gut-based hypnotherapy to improve global symptoms, many patients regularly utilise over-the-counter laxatives and anti-diarrhoeal agents to manage constipation and diarrhoea, often in an episodic manner [1]. It has been suggested that a proportion of patients with IBSM using such medications may be experiencing more severe symptoms, as well as being potentially misclassified due to medication effects on stool form, further adding to diagnostic complexity [23]. Tegaserod, a serotonin receptor agonist that accelerates colonic transit and peristalsis, has been demonstrated to improve symptoms in IBS-M as well as IBS-C, suggesting that the management of underlying constipation may be required in this group despite their varying stool consistency [24]. 

Similarly, a subset of patients experiencing diarrhoea-predominant or mixed symptoms may be experiencing overflow diarrhoea due to severe faecal loading [25]. Bloating, abdominal pain, watery diarrhoea, and faecal incontinence can all be symptoms of obstructive constipation as well as IBS [26]. In this setting, alternative diagnoses such as dyssynergic defecation should be sought through anorectal physiology testing, as these patients may benefit from biofeedback therapy rather than traditional IBS management, regardless of whether they have coexisting IBS [1,27,28]. 

There is therefore a need for biomarkers not only to aid in the diagnosis of FGIDs but also to improve phenotyping, guide individualised and efficacious therapy, and assess response. The utility of objective markers in diagnosis has thus far been largely limited to the exclusion of other disease processes, for example, the use of blood markers including haemoglobin and CRP or invasive tests such as colonoscopy, to rule out inflammatory bowel disease or malignancy [1,29].

Imaging assessment of constipation has largely been limited to abdominal X-rays in a paediatric population [30], though those in adults have produced mixed results: some authors have suggested there exists a similarly wide range of faecal loading within asymptomatic adults [31], whereas others found significant increased colon transit time and faecal loading in an IBS population.

As a real-time, non-invasive imaging technique, gastrointestinal ultrasound (GIUS) has the ability to objectively visualise luminal contents and thus the degree of faecal loading, as well as assessing peristalsis, motility, and underlying inflammation that might be suggestive of IBD [32,33]. GIUS assessment of constipation has been demonstrated to be superior to plain abdominal X-rays with the ability to differentiate between cases and healthy controls, as well as being comparable to computed tomography [34,35]. Furthermore, as its use does not require radiation exposure, GIUS is useful for serial assessment of response to therapy [36,37].

Nevertheless, there are limited data available on the prevalence and severity of objective faecal loading as determined by GIUS in patients with lower FGIDs. It is also currently unknown whether GIUS evaluation of faecal loading correlates with patient-reported symptoms. How useful an assessment of faecal loading may be in guiding targeted therapy has also not been assessed. It is likely, however, that GIUS will be particularly useful in distinguishing those in whom mixed and diarrhoea-dominant phenotypes are in fact secondary to underlying constipation. 

The present study aimed to assess the utility of GIUS use in phenotyping patients with lower FGIDs by comparing objective ultrasound evidence of faecal loading with FGID phenotype based on the Rome criteria. A secondary goal was to assess, in those individuals who had undergone serial GIUS, whether improvement in patient-reported symptoms correlated with an improvement in ultrasound evidence of faecal loading. 

## 2. Materials and Methods

Study participants were recruited from consecutive new patients over the age of 18 years attending the outpatient Functional Gut Clinic at a major tertiary centre between June 2018 and February 2019. The inclusion criteria were a diagnosis of a lower functional gut disorder as defined by the Rome IV criteria (including IBS-D, IBS-M, IBS-C, IBS-U, functional constipation, and functional diarrhoea). These were categorised by a gastroenterologist based on clinical assessment, responses to the Rome IV questionnaire, and exclusion of other pathologies with further investigations when indicated. The exclusion criteria included a diagnosis of coexisting coeliac disease, inflammatory bowel disease, or other organic cause of gastrointestinal pathology. Ethics approval was obtained from The Alfred Health and Monash University Ethics Committee (protocol code 370/18 and date of approval 2 July 2018), and participants provided written informed consent before the commencement of the study to use their de-identified data for research purposes.

### 2.1. Study Protocol 

Participants underwent a clinical assessment by a gastroenterologist followed by a gastrointestinal ultrasound. GIUS was performed by a second experienced gastroenterologist blinded to patient symptoms. Intestinal ultrasounds were performed on a GE LOGIQe or a Canon Aplio i800 machine with both convex (low frequency) and linear (high frequency) probes. Faecal loading was assessed using a modified Leech score [30] where each of five colonic segments (ascending colon, transverse colon, descending colon, sigmoid, and rectum) was scored from 0 to 3 (0 = no faecal loading, 3 = severe) to give a total score out of 15. Longitudinal and transverse views of the colonic segments were used to determine the faecal loading score. Ultrasound features that indicated faecal loading included a distended colon, crescent-shaped acoustic shadows associated with haustrations, or multiple high-echoic spots, as these have been found to be correlated with CT-confirmed faecal loading [35] While this assessment is subjective in nature, it was standardised, as described in Figure 1. As the rectum was not visualised on GIUS in all patients (for example, it may be more challenging to visualise if the patient has an empty bladder), all scores were expressed as a percentage of the total area examined, and the same overall area was used for the first and second ultrasound for each individual participant. Previous preliminary studies performed by the authors have shown that a faecal loading score (FLS) of >37 is consistent with clinically significant constipation, as determined by PAC-SYM assessment, with a sensitivity of 69% and specificity of 75% [34]. Colonic motility, diameter, and small bowel distension were also recorded. 

At the time of the first appointment, participants completed the Rome IV Diagnostic Questionnaire for adult functional GI disorders [38] as well as the symptom severity assessment of multiple GI symptoms seen in IBS (SAGIS) [39] and a 100 mm visual analogue scale (VAS), where 0 indicated no symptoms, and 100 represented the worst symptoms ever experienced. The VAS score evaluated the severity of the number of symptoms, including nausea, abdominal pain, bloating, diarrhoea, constipation, fatigue, and overall gastrointestinal symptoms [40]. 

Based on the clinical assessment by a gastroenterologist and subsequent GIUS, an appropriate evidence-based treatment was recommended during this initial clinic appointment. This included general dietary and psychological measures, as well as pharmacotherapy (such as laxatives, fibre, anti-diarrhoeals if clinically indicated). Participants who underwent baseline GIUS and completed the clinical questionnaires were invited to undertake a second GIUS as well as repeat SAGIS and VAS questionnaires 3–6 months later. 

### 2.2. Endpoints 

The primary endpoint was the correlation between lower FGID subtypes based on Rome IV diagnosis and objective GIUS findings of faecal loading as measured by the modified Leech score, as well as the correlation between predominant symptoms (constipation, diarrhoea, or indeterminant) and GIUS findings of faecal loading. Secondary endpoints included the correlation between the severity of faecal loading and the severity of patient-reported symptoms as measured by the 100 mm VAS. We also assessed the correlation between change in GIUS scores after three to six months of treatment with change in symptom severity as measured by 100 mm VAS. 

### 2.3. Statistical Analysis 

Data were analysed using SPSS Statistics version 28 (IBM Corp: Armonk, NY, USA) and displayed using GraphPad Prism version 9.3.1 for Windows (GraphPad Software: San Diego, CA, USA). The distribution of FLSs was inspected for violation of normal distribution. FLSs for the nine identified diagnoses and three symptom categories were compared using Kruskal–Wallis tests.

The FLS threshold that best discriminated between those with and without indications of constipation, maximising specificity and sensitivity, was identified using receiver operating curve (ROC) analysis. The specificity, sensitivity, positive and negative predictive values, accuracy, and likelihood ratios were calculated with 95% confidence intervals reported. Correlations between FLS and 100 mm VAS scores were calculated using Spearman’s rank correlation coefficient. A *p* value of <0.05 was considered statistically significant.

## 3. Results

### 3.1. Participants 

Eighty-eight patients were recruited for this study and completed an initial clinic appointment, questionnaires, and GIUS. The median age was 46.5 years (range 19–81), and 78% were female (Table 1). 

Fifty-six patients met the Rome IV criteria for IBS: IBS-M = 18, IBS-D = 18, IBS-C = 17, and IBS-U = 3. Twenty-three met the criteria for functional constipation (FC). Four patients had functional diarrhoea (FD). Upon the re-evaluation of full questionnaire responses, five met the Rome IV criteria for alternative diagnoses: two were diagnosed with faecal incontinence (FI), and two had functional abdominal pain syndrome (FAP). One individual was unable to be defined (Table 1).

Sixty-five patients completed a second GIUS during the follow-up period. The remaining 23 patients were either lost to follow-up or completed their second clinic appointment via telehealth due to COVID-19 constraints and were therefore unable to complete an in-person ultrasound assessment. Additionally, 43 of the 65 patients who completed a second GIUS also completed a second 100 mm VAS at the time of GIUS assessment. 

### 3.2. Intestinal Ultrasound Faecal Loading Score by Diagnosis

Those with IBS-D had the lowest median FLS (10.8; interquartile range [IQR] 6.7 to 35.0), closely followed by IBS-U (13.3; *n* = 3, range 8.3 to 53). Medians for other categories from highest to lowest were 40.0 (IQR 20.0 to 53.3) and 40.0 (IQR 10.0 to 60.0) for FC and IBS-C, respectively; 36.7 for FAP, FI, and FD; and 33.3 (IQR 12.1 to 48.3) for IBS-M (Figure 2). There was considerable overlap in FLS between diagnoses, and no statistical difference was observed in FLS between categories (*p* = 0.14 for overall Kruskal–Wallis H).

### 3.3. Intestinal Ultrasound Faecal Loading Score by Symptom Category

When patients were divided into three main symptom-related categories, 40 patients had predominant constipation (IBS-C and FC); 22 had predominant diarrhoea (IBS-D and FD); and for 26 patients, predominant symptoms could not be determined (IBS-M, IBS-U, FAP, and FI). Median (IQR) FLSs were 40.0 (20.0 to 53.3) for constipation-predominant, 13.3 (6.7 to 40.0) for diarrhoea-predominant, and 33.3 (8.3 to 53.3) for indeterminate symptoms (Figure 3; *p* = 0.03). Uncorrected pairwise non-parametric comparisons showed that only differences between constipation and diarrhoea groups were statistically significant (*p* = 0.01).

Of the 40 patients who reported constipation-predominant symptoms, only 24 had evidence of faecal loading on GIUS (FLS > 37), and 16 patients had a normal degree of faecal residue. Of the 22 patients reporting diarrhoea-predominant symptoms, 6 had significant faecal loading evident. Almost half of the patients (41/88) met the FLS criteria for constipation, of whom 24 described constipation-predominant symptoms (IBS-C = 10, FC = 14), 6 described diarrhoea-predominant symptoms (IBS-D = 4, FD = 2), and a further 11 patients had mixed symptoms.

### 3.4. Sensitivity and Specificity of FLC for Predicting Rome IV Diagnosis 

ROC analysis for constipation predominance indicated that the cut point that optimised the sensitivity and specificity of FLC was 36.66 (between >33.3 and ≥40.0). Importantly, these data fit with the previously used FLS of >37 to indicate clinically significant constipation [34].

Based on this FLS cut point for the detection of constipation of 37, the sensitivity of the Rome IV diagnosis of IBS-C or FC for detecting FLS-indicated constipation was 58.5%, while specificity was 66%. Positive and negative predictive values, overall accuracy, and positive and negative likelihood ratios are summarised in Table 2 below. 

Because 26/88 patients had indeterminant symptoms (IBS-M, IBS-U, or Other), which reduced apparent sensitivity and specificity, the analyses were repeated with these patients removed from the analysis. This reduced the number of false negatives for constipation and increased sensitivity to 80% but slightly reduced specificity to 50%.

### 3.5. FLS Correlation with Symptom Severity

There were no statistically significant correlations between the faecal loading score and symptom severity as measured by 100 mm VAS for upper abdominal pain, lower abdominal pain, diarrhoea, constipation, overall gastrointestinal symptoms, and overall quality of life (Appendix A). 

### 3.6. Change in FLS and VAS Scores at Follow-Up 

The mean faecal loading score for the 65 patients who completed a second USS reduced by a mean of 6.48 points from 40.25 to 33.77. The only significant change in the VAS symptom severity score was a reduction in lower abdominal pain by 11 mm (*p* = 0.01).

### 3.7. Change in FLS Correlations with Change in VAS

There were no statistically significant correlations between the change in faecal loading score and the change in VAS scores between the two time points. 

## 4. Discussion

The Rome criteria scaffolds how we think about functional gastrointestinal disorders: how they are diagnosed, classified, and therefore treated and researched. It is reliant on self-assessment by the patient of subjective symptoms, and the overlap between subtypes of FGIDs is increasingly recognised. 

Our study clearly demonstrates that symptoms are a poor predictor of the severity of faecal loading in those with FGIDs. GIUS did correlate with symptoms to a degree: Patients who described diarrhoea-predominant symptoms had significantly lower faecal loading scores compared with those who described constipation (a median FLS of 13.3 in diarrhoea and 40.0 in constipation). This is concordant with previous findings of an FLS of >37 being consistent with a diagnosis of constipation, though it is worth noting that this study also identified a lower mean FLS of 23.6 in a healthy control population [34].

However, there was a wide variance in scores within those describing constipation (IQR 20.0 to 53.3), and sixteen patients described constipation without ultrasound evidence of faecal loading. It is well established that IBS is multi-factorial, and it may be possible that in these patients, with little objective evidence of faecal loading, visceral hypersensitivity and/or psychological factors play a more significant role [41,42]. It may also be possible that in some patients, diet quality, particularly inadequate fibre, may also contribute to a lack of stool bulk on ultrasound despite constipated symptoms. Being able to disentangle these subgroups may allow for more targeted therapy. 

We also identified that those diagnosed with IBS-M had a particularly wide range of objective faecal loading scores (median FLS 33.3 with IQR 12.1 to 48.3). Previous work has suggested that IBS-M is more similar to IBS-C than IBS-D [24,43], and as such, our finding that nearly half of IBS-M patients had an FLS consistent with objective constipation is consistent with this. However, another recent study found that IBS-M patients experienced symptoms more consistent with IBS-D, especially rectal urgency and abdominal pain [23]. It may also simply represent the inherent heterogeneous nature of IBS-M: criteria for IBS-M relies on reporting more than 25% of abnormal bowel movements being constipation and more than 25% diarrhoea [3]. It would therefore be useful to know if these ultrasound findings remained stable over time or changed with predominant symptomatology. This may be an area where GIUS is particularly useful, as repeated measures are possible given the fact that ionising radiation is not required. We are also aware that patients’ normal stool form may be masked by the use of medication to manage symptoms, particularly in this IBS-M subgroup, and this may further confound diagnostic accuracy [23]. 

GIUS clearly showed that a subset of patients describing diarrhoea-predominant symptoms in fact had constipation with overflow diarrhoea being the likely underlying cause: Six patients with a Rome IV criteria diagnosis of functional diarrhoea or IBS-D had an FLS of >37 (including FLSs up to 100 for patients diagnosed with IBS-D). Instead of treating for diarrhoea, it would be crucial to ensure that these patients received appropriate laxative therapy and investigate other coexisting pathology such as dyssynergic defecation. Standard therapy with anti-diarrhoeals would likely considerably worsen their outcomes.

Interestingly, in this study, objective severity of faecal loading as measured by ultrasound was also a poor predictor of reported symptom severity or quality of life, highlighting the complex interplay between physiology and psychosocial overlay. Whilst it may be that we simply did not detect this due to the small sample size, it is well recognised that a major impact of IBS is on quality of life [44,45], and thus symptom assessment remains crucial. If someone has objective faecal loading on ultrasound but is not symptomatic, then there may be little to be gained by pathologising and treating this. However, if the same person presented with constipation and perhaps overflow diarrhoea, it would be clinically useful to distinguish between these two processes using a tool such as GIUS. GIUS may also identify a subset of patients with symptoms of constipation and bloating but no objective faecal loading in whom a different pathophysiological process, such as rectal evacuatory dysfunction or visceral sensitivity, may be driving symptoms. In such individuals, commencing or escalating laxatives is unlikely to be helpful. Tools such as the Bristol Stool Chart are already in routine practice to increase the accuracy and objectiveness of symptom descriptions [46], and GIUS is a natural extension of this. Furthermore, the non-invasive accessible nature of the investigation, which can be undertaken in the clinic, means that therapy can be modified in a contemporaneous manner. The lack of ionising radiation also means the test is repeatable and can be used to monitor response over time. Frustratingly, in this study, we did not see a correlation between a change in symptoms and a change in FLS; however, the number of patients with follow-up ultrasounds was small due to the impact of COVID-19. Treatment in response to the initial intestinal ultrasound result was also clinician-led and not protocolised, which limits the conclusions that can be drawn. 

Crucially, when assessing the overall performance of the Rome IV criteria diagnosis of constipation (IBS-C and FC) in comparison to objective ultrasound findings, the sensitivity and specificity of Rome were low, at 58.5% and 66%, respectively. This means that in our patient cohort, the Rome diagnosis was only slightly better performing than chance in identifying faecal loading, and it performed far below what we would normally expect in medicine for either an effective screening or diagnostic test, being neither highly sensitive nor specific [47,48]. Instead, we propose that FGIDs should be subtyped not only by symptoms but also by the patient’s underlying physiology in order to refine and optimise treatment strategies. Objective biomarkers such as GIUS to define the degree of faecal loading may be an important step towards this and remains a relatively cost-effective point-of-care tool to use bedside in a clinic appointment setting. The next step would involve a larger interventional study where specific standardised interventions based on the degree of faecal loading are implemented.

Our study had several limitations. It was exploratory in nature, and so the sample size was small. Unfortunately, 23 patients did not complete a second follow-up gastrointestinal ultrasound, in part due to COVID-19 impeding face-to-face consultations, and so data about changes in faecal loading score and symptomatology were limited. With these constraints, it is possible that it was not powered to detect findings such as correlations between symptom severity and faecal loading severity. It is acknowledged that the reproducibility of the FLS has not been validated; however, this study is the first study to look beyond just rectal diameter when assessing luminal contents with intestinal ultrasound in patients with lower GI functional disorders. Thus, these results, though preliminary, are important. Further studies are needed to validate these findings. Recruitment was limited to a single tertiary centre clinic in Melbourne and thus may not be relevant to other settings. 

Furthermore, BMI data were not collected during the study. As ultrasound can be less reliable in patients who are overweight or obese, it is possible that this could underestimate the utility of IUS to assess luminal contents. Nevertheless, the study population reflected real-world Australians (patients were not excluded for being overweight or obese), and hence the outcomes are applicable to clinical practice. 

Despite these weaknesses, this study represents an important starting point for further exploration of the role of intestinal ultrasound in FGIDs, and it is the first study comparing symptoms to objective ultrasound findings. It further supports the literature suggesting that IBS-D, IBS-M, IBS-C, FC, and FD exist on a spectrum of diseases with considerable overlap in clinical and now also physiological profiles, and a valid objective assessment would help to delineate these entities further.

## 5. Conclusions

In conclusion, this study demonstrates the inherent unreliability of symptoms in comparison to ultrasound assessment of faecal loading in patients with FGIDs. Despite this, symptoms routinely underpin clinical diagnosis, the Rome criteria classification of subtypes, and therefore management. These findings should prompt further a shift towards the use of additional objective physiological assessments of patients with FGIDs. The non-invasive and radiation-free approach of intestinal ultrasound makes it an ideal modality to objectively assess this patient group.

## Figures and Tables

**Figure 1 jcm-13-04759-f001:**
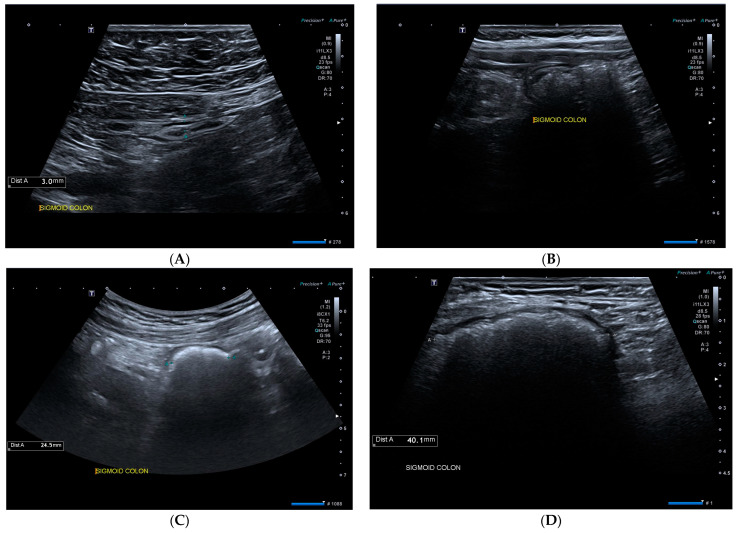
Sonographic images of faecal loading with corresponding scoring criteria: (**A**) empty colon (score = 0)—no stool within lumen, colon collapsed; (**B**) mild loading (score = 1)—some stool within lumen but colon remains relatively collapsed; (**C**) moderate loading (score = 2)—stool within lumen, colon full, rounded appearance, and unable to see the posterior wall; (**D**) severe loading (score = 3)—stool with lumen, colon distended, unable to see the posterior wall, and other features such as crescent-shaped acoustic shadows with haustrations.

**Figure 2 jcm-13-04759-f002:**
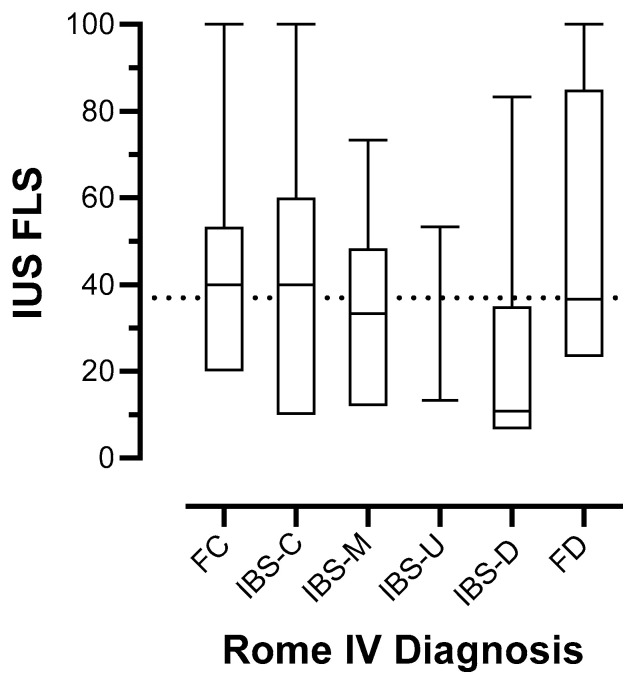
Distribution of intestinal ultrasound faecal loading scores (IUS FLSs) stratified by lower functional gastrointestinal disorder as defined by the Rome IV criteria (functional constipation; irritable bowel syndrome with predominant constipation; IBS-mixed; IBS indeterminant; IBS with diarrhoea; functional diarrhoea). Five patients categorised as “Other” are removed; *n* = 83.

**Figure 3 jcm-13-04759-f003:**
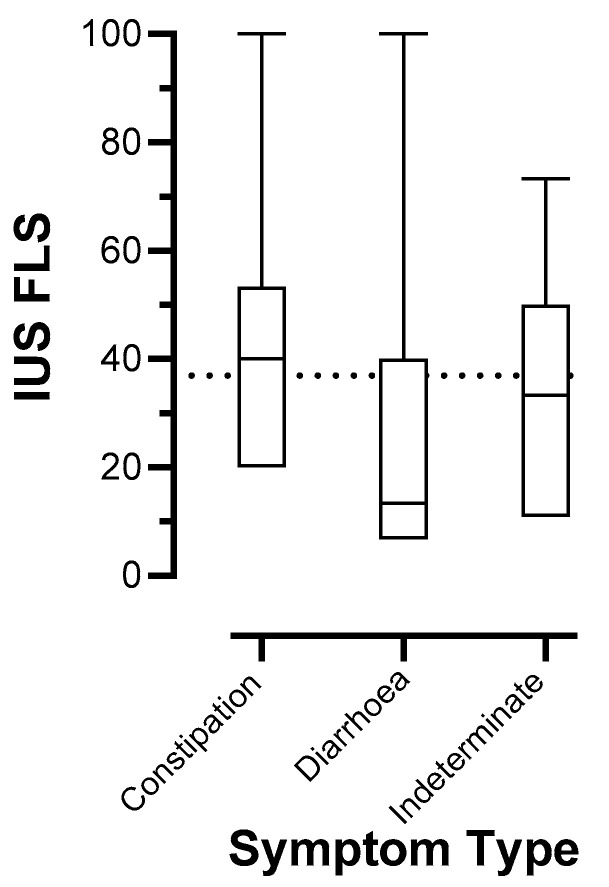
Distribution of intestinal ultrasound faecal loading scores (FLSs) stratified by predominant symptom category. Five patients categorised as “Other” are removed; *n* = 83.

**Table 1 jcm-13-04759-t001:** Patient demographics and FGID diagnosis based on the Rome IV criteria.

	*n* (%)
Gender	
Male	19 (21.5)
Female	69 (78.4)
Median age (years)	46.5
Age 18–29	19 (21.6)
Age 30–49	31 (35.2)
Age 50+	38 (43.2)
FGID diagnosis based on Rome IV criteria	
IBS-C	17 (19.3)
IBS-M	18 (20.5)
IBS-D	18 (20.5)
IBS-U	3 (3.4)
Functional constipation	23 (26.1)
Functional diarrhoea	4 (4.5)
Other	5 (5.7)

Notes: Other includes faecal incontinence (*n* = 2), functional abdominal pain (*n* = 2), and unknown (*n* = 1).

**Table 2 jcm-13-04759-t002:** Features of the Rome IV criteria for diagnosis of constipation (as diagnosed by FLS > 37).

Sensitivity (95% CI), %	58.5 (48.2 to 68.8)
Specificity (95% CI), %	66.0 (56.1 to 75.9)
Positive Likelihood Ratio (95% CI)	1.72 (1.07 to 2.76)
Negative Likelihood Ratio	0.63 (0.41 to 0.95)
Positive Predictive Value (95% CI), %	60.0 (49.8 to 70.2)
Negative Predictive Value (95% CI), %	64.6 (54.6 to 74.6)
Accuracy (95% CI)	62.5 (52.4 to 72.6)

## Data Availability

De-identified data are available upon request from the corresponding author.

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
