# Peer review of "Moving beyond Symptom Criteria to Diagnose and Treat Functional Disorders: Patient-Reported Symptoms of Functional Lower Gastrointestinal Disorders Correlate Poorly with Objective Assessment of Luminal Contents Seen on Intestinal Ultrasound"

_jcm, 2024, doi:10.3390/jcm13164759_

Round 1

Reviewer 1 Report

Comments and Suggestions for Authors

The study aims to evaluate the usefulness of gastrointestinal ultrasound in establishing the quantity of feces (fecal loading score: FLS) present in the large intestine (evaluating the 5 segments into which it can be divided: ascending colon, transverse colon, descending colon, sigmoid colon). and rectum) in patients with functional disorders of the lower gastrointestinal tract according to the Rome IV criteria (IBS-D, IBS-C, IBS-M, IBS-U, Functional diarrhea (FD) and Functional Constipation (FC). The assessment is then correlated with the symptoms reported by the patients. The authors state that

1)the diagnosis of the various subtypes of functional disorder of the lower gastrointestinal tract based on symptoms are poorly correlate with LFS.

2)having an evaluation of the FLS could therefore be useful for a better diagnostic-therapeutic definition in this kind of patients.

However in this paper:

- The methodology used for the evaluation of FLS is not clear. The use of a modified Leech score is reported, but the Leech score is done using Xray methods: which ultrasound parameters were used in this case to establish the quantity of feces present in the colon?

- Furthermore, the evaluation remains very subjective (grade 0-3) and its reproducibility is completely to be validated

- It is not clear why FLS value greater than 37 was used as an indicator of clinically significant constipation

In conclusion the paper has the merit to hypothesize the use of a non invasive method to evaluate the bowel transit but  the authors should better verify the reliability of the suggested method and particularly the interobserver variations

Author Response

Queries raised by Reviewer 1

Thank you for your time reviewing our paper. Please see responses to your queries below. 

1)  The methodology used for the evaluation of FLS is not clear. The use of a modified Leech score is reported, but the Leech score is done using Xray methods: which ultrasound parameters were used in this case to establish the quantity of feces present in the colon?

Similarly to the Leech score, the scoring 0 – 3 was subjective in nature but was scored by a gastroenterologist ultrasonographer blinded to patient presentation and symptoms. Features of significant faecal loading detected on ultrasound included a distended colon with marked haustral pattern, crescent shaped acoustic shadow or multiple high echoic spots as these findings have previously been shown to correlated with faecal loading as seen on CT.

Further detail has been added to the text (see below) and the following reference applied.

“GIUS was performed by a second experienced gastroenterologist blinded to patient symptoms. Intestinal ultrasounds were performed on a GE LOGIQe or a Canon Aplio i800 machine with both convex (low frequency) and linear (high frequency) probes. Faecal loading was assessed using a modified Leech score30 where each of five colonic segments (ascending colon, transverse colon, descending colon, sigmoid, rectum) was scored from zero to three (zero = no faecal loading, 3 = severe) to give a total score out of 15. Longitudinal and transverse views of the colonic segments were used to determine the faecal loading score.  Ultrasound features that indicated faecal loading included a distended colon, crescent-shaped acoustic shadows associated with haustrations or multiple high echoic spots as these have found to correlated with CT confirmed faecal loading38 

Yabunaka, K.; Matsuo, J.; Hara, A.; Takii, M.; Nakagami, G.; Gotanda, T.; Nishimura, G.; Sanada, H. Sonographic Visualization of Fecal Loading in Adults: Comparison With Computed Tomography. Journal of Diagnostic Medical Sonography 2015, 31 (2), 86-92. DOI: 10.1177/8756479314566045).

2) Furthermore, the evaluation remains very subjective (grade 0-3) and its reproducibility is completely to be validated.

It is acknowledged that the FLS is a subjective score which has not been validated we agree the reproducibility of IUS in assessing luminal contents has not been confirmed  

As such, the text has been altered to highlight these limitations.

“It is acknowledged that the reproducibility of the FLS has not been validated however this study is the first study to look beyond rectal diameter when assessing luminal contents with intestinal ultrasound in patient with lower GI Functional disorders. Thus, these results, though preliminary, are important”.

3) It is not clear why FLS value greater than 37 was used as an indicator of clinically significant constipation.

This value was obtained from preliminary work done by our unit (published in abstract form and referenced within the paper à Luber RP, T. K., Gerstenmaier J, Friedman AB, Asthana A, Gibson PR, Burgell RE. . Superior performance of gastrointestinal ultrasound over abdominal x-ray in the assessment of constipation. Gastroenterology,2019 May 1;156(6):S-594.

This study of 17 healthy individuals and 69 patients with chronic constipation showed the superiority of IUS over AXR in the assessment of constipation. ROC analysis of GIUS showed good differentiation of patients and controls, with an AUC of 0.78 (95% CI: 0.66-0.89. A FLS cut off score of 37 had a sensitivity and specificity of 67% and 75% respectively.

 To address the reviewer concerns the following line has been added to the paper

Previous preliminary studies performed by the authors have shown that a faecal loading score (FLS) of >37 is consistent with clinically significant constipation as determine by PAC-SYM assessment with a sensitivity of 69% and specificity of 75%34.

Reviewer 2 Report

Comments and Suggestions for Authors

I have reviewed in detail the paper entitled: “Patient reported symptoms of lower functional gastrointestinal disorders correlate poorly with degree of fecal loading seen on intestinal ultrasound”. In this article, the authors evaluated the utility of GIUS use in phenotyping patients with lower FGID by comparing objective ultrasound evidence of fecal loading with FGID phenotype based on the Rome criteria. I comment the following:

I think the title sounds like an objective, it would be appropriate to modify it.

The introduction is adequate, with the necessary information for readers.

I think you should include basic information (some specifications) of the equipment you used for the ultrasound.

I would greatly support including ultrasound images to better appreciate the differences that you mentioned in the text.

Did you evaluate the BMI of the patients? What percentage of them were overweight or obese? This is a relevant question because the utility of GIUS in the setting of obesity can be limited, where the depth of penetration may impede accuracy of imaging as well as the capacity for color Doppler ultrasound. Studies have estimated that the prevalence of obesity in these conditions is close to 40%. 

Author Response

Response to review 2.  

Thank you for your time in reviewing our paper. Please see reponses to your queries below. 

  • I think the title sounds like an objective, it would be appropriate to modify it.

The title has been modified as below

“Moving beyond symptom criteria to diagnose and treat functional disorders: Patient reported symptoms of functional lower gastrointestinal disorders correlate poorly with objective assessment of luminal contents seen on intestinal ultrasound”.

  • The introduction is adequate, with the necessary information for readers.

Thank you

  • I think you should include basic information (some specifications) of the equipment you used for the ultrasound.

Thank you for drawing our attention to this oversight. This has been corrected and the following added to the methods section.

 “GIUS was performed by a second experienced gastroenterologist blinded to patient symptoms. Intestinal ultrasounds were performed on a GE LOGIQe or a Canon Aplio i800 machine with both convex (low frequency) and linear (high frequency) probes. Faecal loading was assessed using a modified Leech score30 where each of five colonic segments (ascending colon, transverse colon, descending colon, sigmoid, rectum) was scored from zero to three (zero = no faecal loading, 3 = severe) to give a total score out of 15. Longitudinal and transverse views of the colonic segments were used to determine the faecal loading score.  Ultrasound features that indicated faecal loading included a distended colon, crescent-shaped acoustic shadows associated with haustrations or multiple high echoic spots as these have found to correlated with CT confirmed faecal loading38 

  • I would greatly support including ultrasound images to better appreciate the differences that you mentioned in the text.

 Thank you for these suggestions. These have been added to the text – see figure 1

  • Did you evaluate the BMI of the patients? What percentage of them were overweight or obese? This is a relevant question because the utility of GIUS in the setting of obesity can be limited, where the depth of penetration may impede accuracy of imaging as well as the capacity for color Doppler ultrasound. Studies have estimated that the prevalence of obesity in these conditions is close to 40%. 

Unfortunately, we did not collect data on BMI during the study. We acknowledge this is a limitation, however as 1/3 of Australians are overweight or obese it is likely to result in an underestimation rather than overestimation of the utility of the investigation modality. The following line has been added from line 292

“Furthermore, BMI data was not collected during the study. As ultrasound can be less reliable in patients who are overweight or obese it is possible that this could underestimate the utility of IUS to assess luminal contents.  Nevertheless, the study population reflected real world Australians (patients were not excluded for being overweight or obese) and hence the outcomes are applicable to clinical practice”.  

Round 2

Reviewer 1 Report

Comments and Suggestions for Authors

There is an issue which remains to be elucidated: how the authors evaluate the amount of feces present in the colon: in my opinion this evaluation is highly
subjective (grade 0-3) and its reproducibility is completely to be
validated

Author Response

Responds to reviewer.

Point 1: There is an issue which remains to be elucidated: how the authors evaluate the amount of feces present in the colon: in my opinion this evaluation is highly subjective (grade 0-3) and its reproducibility is completely to be validated. 

We agree (and have acknowledged in the discussion) that the modified Leech Score is a subjective assessment and it has not been validated. However this is similar to the current available methods of assessing faecal loading using abdominal Xray. Whilst the ultrasound assessment is subjective, the approach utilised in the study was standardised. To highlight this, we have now included further description and images of the criteria used for each scoring value (see figure 1).